# A Survey of Digital Television Interactivity Technologies

**DOI:** 10.3390/s22176542

**Published:** 2022-08-30

**Authors:** Volnei da Silva Klehm, Rodrigo de Souza Braga, Vicente Ferreira de Lucena

**Affiliations:** 1SIDIA R&D Institute, Manaus 69055-035, Brazil; 2UFAM-CETELI, PPGEE, and PPGI, Federal University of Amazonas, Manaus 69067-005, Brazil

**Keywords:** digital TV, interactive systems, middleware, TV broadcasting

## Abstract

This paper presents a survey of the worldwide use of Digital Television interactivity (DTVi) standards. First, we recall some concepts of first-generation interactivity middlewares released in the early 2000s, such as the European MHP (based on Java) and the Japanese BML (based on XML). Then, we cover the new standards (emerging after 2010) that introduced the new Integrated Broadcast Broadband (IBB) model, which combines broadcast signals with a broadband interface and leverages synergies to offer high-quality, flexible, interactive and customized services and applications to viewers. Regarding IBB systems, we also cover the main aspects of their architectures and innovations introduced by this kind of technology, such as support to companion devices, e.g., smartphones, tablets, targeted advertisements, and integration with Internet-of-Things (IoT) devices. Finally, we show the current adoption of different IBB systems around the world as well as current challenges regarding IBB technologies.

## 1. Introduction

Besides having higher image definition and better audio quality, one of the key new features of digital television compared to its analog predecessors is the ability to execute applications embedded in the broadcaster data stream. This feature has moved the viewer from a simple passive role to a more active one. People at home can participate actively in shows by voting, sending messages to the TV presenter, deciding who will be eliminated on reality shows, buying products advertised, or receiving customized advertisements based on personal preferences.

TV interactivity is not a new concept. For example, in the 1970s, a service called Teletext was launched in the United Kingdom, allowing the BBC to display text for viewers containing information related to news, disaster alerts, daily weather, closed weather captioning, etc. Teletext transmitted data using the Vertical Blanking Interval (VBI) between video lines. Additionally, in the 1970s, Warner introduced a system called Qube. It consisted of a set-top box that allowed viewers to access the TV programming, select channels, purchase movies (pay-per-view) and respond to interactive programming [1]. Other European countries and even the United States adopted similar technologies in the following years [2]. However, these first initiatives had to face some obstacles, e.g., the high costs of having two-way networks installed in viewers’ homes at that time.

In the following decades, significant technological advances enabled the emergence of several proprietary interactive solutions (e.g., MediaHighway, Liberate, Microsoft TV, NDS Core, Betanova, and OpenTV) [3]. Then, in the late 2000s, TV standardization bodies worldwide proposed the first open standards, from which emerged standards such as Multimedia Home Platform (MHP), Broadcast Markup Language (BML), and Advanced Common Application Platform (ACAP) [4]. In addition, in the past decade (the 2010s), these systems have evolved to enable the creation of services that integrate broadcast and broadband channels in the best way possible. These new middlewares are intuitively called Integrated Broadcast Broadband (IBB) systems.

This article provides a starting point for anyone who wishes to know about interactivity systems for TV digital worldwide and covers topics such as the evolution of the standard and its current development and adoption state. By reading this, we expect the following questions: “Where is such standard used?”, “Why it has been adopted?”, “What are its main characteristics?” and “How can we compare standards with each other?” to be answered. This article aims to present the current state of the art regarding television interactivity and current challenges, and to focus on describing some IBB system particularities.

This article is organized as follows: Section 2 provides background regarding legacy standards. Section 3 focuses on the Integrated Broadcast Broadband (IBB) standard. Section 4 shares the open challenges to the IBB standard. The conclusion closes the paper.

## 2. Early ITV Open Standards

Until the mid to late 2000s, proprietary middlewares, such as OpenTV, PowerTV, and Microsoft, dominated interactivity TVs. In other words, the services and applications on these proprietary middlewares are tied to the platforms. Furthermore, the owners of these middlewares control the specifications of the set-top boxes used in these networks and the application formats. Consequently, a fragmented market was created, demanding a high cost to deploy the same service in this heterogeneous environment [4], which resulted in the development of open TV interactivity standards around the globe. Open standards can bring many benefits; for example, they guarantee that compliant systems will work together, no matter which manufacturer provides the equipment.

Several open standards have been proposed in several countries, and their main specifications are:The **Multimedia Home Platform** (MHP): designed by the Digital Video Broadcasting (DVB) project and mainly used in Europe;The **Digital TV Application Software Environment** (DASE): designed by the Advanced Television Systems Committee (ATSC) in the United States;The **Broadcast Markup Language** (BML): designed by the Association of Radio Industries and Businesses (ARIB) in Japan;The **Ginga**: designed in Latin American countries and developed in Brazil.

In 1997, the DVB project (a consortium of the world’s leading media companies) developed the first version of the Multimedia Home Platform (MHP) specification, which enabled interoperable applications to be downloaded from DVB broadcasters’ networks and executed on receivers from any manufacturer [5]. MHP was released as a standard by the European Telecommunications Standards Institute (ETSI) in January 2000 and was updated annually; its final revision was the MHP 1.1.2, completed in 2005. This version includes all functionalities from all MHP profiles [5].

MHP provides an execution environment based on a Java Virtual Machine (JVM) and generic APIs to access the interactive digital TV receiver hardware. It supports applications written in either Java or HTML (XHTML 1.1, and includes CSS 2.0, DOM 2.0, and ECMAScript) [6]. Java applications for MHP receivers are called DVB-J applications. Similarly, HTML applications are known as DVB-HTML applications. However, Java is the most used programming language by TV broadcasters.

DVB-J applications use the Java Xlets (*javax.tv.xlet* package, an abstract concept similar to applets for Web pages that Sun Microsystems introduced in the JavaTV specification. Like applets, the Xlet interface allows an external source (the application manager in the case of an MHP receiver) to start and stop an application [5].

One of the main variants of MHP was the OpenCable Applications Platform (OCAP), published by CableLabs in December 2001 [7]. It used the MHP standard to develop a common platform to run services and applications on any cable TV system in the North American market, regardless of the set-top box or television receiver hardware.

The Global Executable MHP (GEM) specification was created to allow the use of MHP in non-DVB networks. It defined a subset of MHP, which removed the transmission-related elements of the MHP specification but retained the application APIs, thus allowing broad content compatibility across a range of new delivery platform developments. In addition, it included interoperability across different middleware specifications, e.g., MHP, Blu-ray, OCAP, and Advanced Common Application Platform (ACAP).

MHP was the basis for several interactivity technologies for TV being used especially in Europe. By supporting web technologies as well as Java Xlets, it became a reference system for all systems that came after it. Using these technologies, MHP reached a great portability as well as consistency (same experience in different devices) by adopting Java. It opened a door for multiplatform compatibility as the application may run on a Java VM instead of directly on a platform, allowing the same binary to be executed on different types of devices. MHP was a great starting point for interactive television. However it is now being replaced by HbbTV, which will be explained in more detail in Section 3.4.

### 2.1. The DASE Standard

ATSC released the DTV Application Software Environment (DASE) middleware in 2003 as an A/100 specification [8]. Its first version was restricted only to local interactivity, as it did not specify a return channel to provide communication from the TV terminal to the broadcaster. Similar to the MHP standard, it supports declarative and procedural applications as show in Table 1. The declarative application is defined upon web technologies (i.e., XHTML, CSS, and ECMAScript). The procedural application is Java TV Xlet, composed of Java-compiled files and resource contents such as video and audio files.

DASE was used as the basis for the next-generation Advanced Common Application Platform (ACAP) standard, released by ATSC two years later. It was idealized as a standard that allowed retro-compatibility with existing DTV applications already deployed to previous systems OCAP and DASE in the United States market. Canada and Korea also adopted ATSC standards, including ACAP, for their terrestrial transmission services [8].

The architecture of the ACAP standard is similar to the MHP standard [4], as it has two execution environments: ACAP-J (Java TV Xlet applications) and ACAP-X (XHTML applications). These environments can request resources from the standard content-decoder module, which handles the resources (e.g., images, font, ZIP, etc.). Applications running in the ACAP-J environment can be written in Java Xlet, with the following extension API: closed caption, Service Information (SI), user interface event handler, and Document Object Model (DOM) integration (a bridge between procedural and declarative environment). ACAP-X application is a multimedia document composed of the XHTML markup language that can run on a web browser. In addition, there is a security framework where ACAP applications may request access to privileged operations of both declarative and procedural environments. For example, if an application needs to handle navigation cookies, it should first grant this access through a security framework.

Despite its limitation, DASE is the equivalent to several middleware specifications around the world. The fact that it supported Java Xlet and web applications was undoubtedly an advantage when compared to Ginga, which took a different road when choosing programming languages that are not popular, like NCL. At that time, DASE offered several advantages, especially when compared to previous analog systems (only closed caption data available on NTSC). However, with the advance in telecommunication, new systems had to be implemented, and full interactivity became a must-have feature as broadband communication systems emerged. DASE is a foundation stone present on newer ATSC middleware specifications, and that is its legacy.

### 2.2. The BML Standard

The Broadcast Markup Language (BML) is an XML-based standard created by the Association of Radio Industries and Businesses (ARIB) and adopted as the Japanese standard STD B-24 (Data Coding and Transmission Specification for Digital Broadcasting). The BML development happened in 1999. The standard was initially developed for satellite communications but allowed for terrestrial digital TV broadcasting when its first version was released. BML has XHTML, CSS 2, ECMAScript in its base specification. It is derived from an early draft of XHTML 1.0 strict. Some subset of CSS 1 and 2 is supported, as is ECMAScript [9].

Compared to other standards, the classification of interactive applications on ARIB STD B-24 is obscure, since it only specifies BML and ECMAScript as declarative and procedural languages, and only textual applications were allowed [10]. On the other hand, on standards such as MHP or ACAP, binary formats are also allowed, and specific virtual machine implementations execute Xlet applications.

Even taking a different way from other middleware solutions when not supporting binary formats such as Java Xlets, BML is still very interesting and easy to support because it relies on W3C standards such as XHTML1.0, CSS, and DOM for dynamic presentation changes [11], which guarantees a series of benefits including native support on several browsers. This fact has created an advantage when compared to systems that decided to take a different approach and implement their own solutions such as Ginga, which, besides being similar to BML in the sense that is also mainly based on XML-based language, in this case Nested Context Language (NCL) instead of XHTML, lacks native support on web engines and requires more effort to be developed and maintained, as discussed later in this article.

### 2.3. The Ginga Standard

Ginga is the given name of the middleware specification for the Brazilian terrestrial digital television interactive system [12]. The Brazilian Association of Technical Standards (ABNT) standardized and adopted it in 2009 [13]. Later, it became an ITU recommendation as ITU-T H.761 for IPTV services [14].

The Brazilian Digital Television Standard is based on Japan’s Integrated Services Digital Broadcasting (ISDB) standard. Initially, the Brazilian standard also supported a procedural Java-based middleware known as Ginga-J. However, its support became optional in 2016 in favor of Ginga-NCL, which is royalty-free and straightforward to develop. The Nested Context Language (NCL) is an XML-based declarative language adopted by Ginga middleware. Like other declarative languages, it emphasizes a high-level description rather than an algorithmic implementation.

The NCL language model not only aims at declarative support for user interaction, but it also aims at declarative support for content adaptations and content presentation forms; declarative support for multiple display devices, or even in a wider area, editing/production of the application at exhibition time (live). For some cases, such as when dynamic content generation is required, Lua scripting language [15] can be used to provide the NCL document with such ability. NCL is similar to SMIL language [16]. However, it can also deal with spatial and temporal synchronism in its most general form, treating user interaction as a particular case [17].

Besides enabling dynamic live content, Lua specification for Ginga also provides access to low-level features available on an interactivity terminal to be integrated into the application, such as return channel interfaces such as the TCP and SMS protocols, allowing a bi-directional interactivity experience with the possibility to send data back to the broadcaster [13].

Ginga has several profiles, depending on the version and the type of platform it intends to be executed. Below follows a description of each profile [18]:*Ginga-A:*Includes all basic middleware functionalities, such as NCL, Lua, Lua Specific APIs, basic Video/Audio Decoding, and image decoding;*Ginga-B:*Besides all features available on Profile A, Profile B adds MPEG-1 video decoding as well as some specific features for mobile terminals, e.g., SMS support as a return channel;*Ginga-C:*Profile C adds a catalog user interface (AppCatUI), private bases, support for MPEG-4 containers, HTTP, and live editing protocols. Ginga-C was considered an IBB system (discussed in Section 3.3); however, it is still missing multiple screens support, added to new specifications in 2018.*Ginga-D:*On Ginga-D, several media formats and several streaming formats have been added. Besides that, now the platform supports HTML5 applications and previous versions of the NCL language (see reference [18]).

Most of the current developments regarding Ginga are related to fully complying with Ginga-C and Ginga-D profiles; however, due to the fact that Ginga uses NCL language, which is not a popular choice of programming language between developers, some hybrid solutions have been proposed in order to allow compatibility with the HbbTV application and services [18]. Other current developments are focused on creating solutions that are fully platform-independent and able to be executed on different devices such as the work in [19], which presents a fully portable JavaScript implementation of Ginga-NCL, thanks to technologies such as WebAssembly, supporting the Lua language interpreter in the Web context, making it easier to embed it to new devices and create authoring tools as shown in [20].

Besides all these developments, it is not clear what the future holds for Ginga. Vendors in Brazil currently have tax benefits by factoring devices compatible with Ginga specifications [21,22] at the same time as broadcasters not having too much interest in developing Ginga interactive content. Currently it is rare to find such an application on a daily TV program. Another problem regarding Ginga is fragmentation: Brazil has a well-documented standard [13]; however, there is no organization to certify what a Ginga implementation is according to this specification, and different vendors’ solutions may behave in slightly different ways, making complex interactive application development difficult as broadcasters may have to consider several platform nuances when creating interactive content.

## 3. Integrated Broadcast Broadband (IBB)

In the previous sections, we have discussed the first generation of interactivity technologies available worldwide. Although these technologies were very innovative when they first debuted, they lacked features found in modern TVs, such as a broadband connection to the Internet, support for third-party software installation, and integration with other devices.

### 3.1. Limitations of the Legacy Standards

One of the problems regarding adopting legacy standards seems to be related to manufacturer standardization. For example, as mentioned in Section 2.3, in Brazil, middleware implementation is fragmented, no organization certifies if a middleware implementation is correct, and, as a result, functional consistency between different devices is hard to achieve.

Furthermore, standards such as Ginga or BML use programming languages that are not popular across the developer community [9,14,23], making it difficult to find programmers for these systems. Newer IBB systems adopt the HTML5 standard, which is popular among developers worldwide [18,24,25]. On the other hand, Java-based standards, such as MHP, have a vast community of developers; however, they use complex licensing models, resulting in the Ginga-J standard not being mandatory in Brazil anymore. Another possible reason that legacy solutions did not gain more developers over time is the limited set of tools available to develop interactive applications.

Finally, the advancement of technology of current multimedia systems, which are complete computer systems, allows application and market integration, which was not present on TV and set-top-boxes when these legacy standards were specified. The legacy solutions described previously are still supported. However, they targeted devices with minimal processing power and connectivity capabilities. Due to this fact, new approaches have been proposed and standardized in a new class of middleware named Integrated Broadcast Broadband (IBB) that can take advantage of broadcast and broadband in a way that the legacy ones cannot. The development of IBB standards was possible due to the exponential growth, in the 2000s, of broadband networks, with high speeds and reliability [26].

### 3.2. Consequences of the Popularity of Broadband Networks

With the increase in broadband speeds, consumer behavior towards media content consumption has changed dramatically. End users began to consume a lot of new online services, such as video on demand and social media. With an eye on this new consumer behavior, TV manufacturers developed connected TV (or smart TV), products that enabled users to consume services delivered over broadband networks, e.g., YouTube, Netflix, and others.

Although these connected televisions provide many advantages, they do not guarantee fully interactive experiences. For example, all connected TV has two communication interfaces: one for the TV tuner (broadcast) and one for the Internet connection (broadband). However, these interfaces do not communicate with each other. Hence, when a user is watching a TV program, the broadband connection is not used, and vice versa. Moreover, when any smart TV application is engaged, the TV signal is disabled. Thus, we argue that a connected TV is not a convergent device. Instead, it is a multi-purpose device that enables the viewing of broadcast television programming or the use of discrete and restricted add-on functionalities through an Internet connection.

Thus, new middlewares have emerged worldwide to create a solution to this limitation, enabling better integration between broadband and broadcast interfaces. In addition, they allow the exploration of opportunities to drive user engagement and maximize end-user entertainment by offering a range of new services.

### 3.3. IBB System Concepts

Integrated Broadcast-Broadband (IBB) systems combine technologies from broadcast and broadband. The main difference from “smart TV” (also connected to the Internet) is that it can show online broadband information related and synchronized with the broadband content [27]. Different IBB standards are in use or development around the world. Table 2 presents the IBB systems deployed around the world and their features, where we have: HbbTV in Europe and some countries in Asia and Oceania, HybridCast in Japan, ATSC 3.0 in the United States, and DTV Play in Brazil. In the following sections, we will describe them, showing everything from their receiver architecture to the most recent related works.

The ITU-T J.205 recommendation (requirements for an application control framework using integrated broadcast and broadband digital television) [28] specifies the general requirements for IBB systems, such as the system model, architecture, and behavior of applications and services. In addition, this document describes three important entities in IBB systems shown in Figure 1: the Broadcaster, the Application Repositories, and the Receiver:*Broadcaster:*The broadcaster offers service providers digital broadcasting signals, metadata, and video content. In addition, it can also provide application control and authorization information, e.g., allowing the start of available apps and permission changing to access broadcast resources. The broadcasting signal can transmit all this information.*Application Repository:*The Application Repositories are responsible for distributing applications for on-demand installation. Specifically, it creates and distributes content and applications that enable the provision of services. It can also manage the servers that allow such applications.*Receiver:*The receiver executes IBB apps, controls display via the application, interacts with users and other devices, and receives and presents linear broadcasting content. The receiver implements industry-standard functions and APIs that enable programs to be executed. In addition, the receiver must have the capability to connect simultaneously to the broadcast and broadband networks. The broadcast network allows the transmission of TV signals, such as terrestrial, satellite, Internet, or cable. For in-depth understanding of these transmission techniques, see [29]. Via this connection, it can receive the standard broadcast audio/video, application data, and application signaling information. On the other hand, the broadband network allows two-way communication with the application provider and receives non-synchronous content, such as video on demand or applications.

Furthermore, by the ITU-T J.205 recommendation [28], the developers of IBB systems are also required to take into consideration the following:Backward compatibility with current broadcast systems:–IBB systems must have backward compatibility with current DTV systems. These systems should be designed to allow the coexistence of IBB applications with conventional broadcast applications, and they should be transparent to the end user when navigating through DTV services.User privacy and data protection:
–IBB system also must ensure that user privacy and user data are protected. To protect user data, it should define a policy on which kind of application has the privileges to access the Internet and sensible information stored on the IBB receiver.Application management:
–The IBB system must allow the IBB DTV service provider to control the associated IBB applications’ execution, availability, and visibility. Additionally, the end user should be able to open an application catalog and launch a stand-alone application on TV while allowing broadcasters to trigger events or kill an application.Types of services or applications:
–The recommendation ITU-T J.205 [28] assumes that there are two types of IBB applications:
∗*Stand-alone IBB applications:* Applications that are not available through the DTV service. The end user would manually launch them through the application catalog user interface.∗*Service associated IBB applications:* Applications that are part of the IBB DTV services. They are delivered as part of the DTV service.

### 3.4. HbbTV

The first version of the HbbTV was published by the European Telecommunications Standards Institute (ETSI) as specification TS 102796 in June 2010 [30]. It was composed of three standards: Consumer Electronics—HyperText Markup Language (CE-HTML) browser, the Open Internet Protocol Television Forum’s (OIPF) Browser profile, and the Digital Video Broadcasting (DVB) Signaling and Transport.

The CE-HTML specification defines HbbTV’s browser functionality [31]. It is based on W3C Web standards and defines an HTML profile. It supports applications written in XHTML, CSS, and JavaScript. Furthermore, it provides a JavaScript interface to handle some TV functionalities (e.g., screen resize and channel change). In addition, HbbTV’s browser supports Asynchronous JavaScript and XML (AJAX) requests, allowing applications to request data from a server and update Web pages without reloading. In this way, application developers can create HTML apps that look and feel similar to modern Web services. As a result, TV integration efforts can be minimized to optimize various viewing scenarios.

The OIPF browser standard, previously designed for DVB IPTV systems, provides an interface to the DVB world. APIs that implement the OIPF browser specification can be used to combine TV content with HTML pages, which adds significant events to the timer list, radio selection, or DVB television services, and allows the reading of DVB data.

In 2016, the ETSI released the HbbTV v2 specification, which introduced some new components, such as HTML5 and Companion Screen, to offer a better experience to connected TVs. Figure 2 depicts the overall system architecture of HbbTV v2. The HbbTV terminal receives DTV signal through DVB transmission (e.g., DVB-T for cable, DVB-S for satellite and DVB-C for cable broadcasting), where it receives AIT data, linear A/V content, non-linear A/V content, application data, and stream events. In addition, the broadband interface can process Internet connections to request application data from servers and non-linear A/V content. An essential mechanism of any IBB system is the synchronization manager, allowing the synchronization of non-linear content delivered through the Internet to contents received via the broadcast interface.

The application manager entity manages the interactive application, where it is responsible for evaluating the AIT events to control the application lifecycle. This way, it can determine when the application should be presented or not in the HTML5 browser environment. The HTML5 browser supports new features, such as CSS3, DOM3, Canvas 2D, video elements, and Web Sockets. In contrast to HbbTV v1.0/1.5 which used HTML4, CSS and DOM Level 2.

Another important feature introduced by HbbTV v2 was the Companion Screen interface, enabling the HbbTV terminal to discover other devices and be discovered by Companion Screen devices. Through this interface, an interactive application running in the HTML5 browser can request an application to be installed or started on a Companion Screen device and vice versa. This connection between devices brings opportunities to improve interactive services, such as facilitating access to synchronized audio descriptions, subtitles, clean audio, or sign language video through the second screen [32].

And finally, in 2020, ETSI released a specification for targeted advertisements (TA Specification TS 103 736-1 V1.1.1) [33]. It enables the delivery of digital advertisements over broadband, replacing part of the linear TV services received by the broadcast interface. As presented in Figure 3, two users with distinct preferences can receive personalized advertisements from the ad server. This behavior is controlled by the HbbTV Digital Ad Substitution (DAS) application, which requires that HbbTV terminals implement a mechanism that enables advertising content to be pre-fetched, buffered, and played. The process begins with the broadcaster signaling that the *placement opportunity* is near. Next, the receiver asks the ad server for an advertisement that can be played. Then, the receiver preloads the ad. Once it is ready and the right moment is reached, the receiver switches from the broadcast to the advertisement. After an ad is finished, the receiver switches back to the broadcast.

Some efforts aim to facilitate integration with Internet-of-Things (IoT) and smart-home systems. For example, the work in [34] proposed a framework for the seamless integration of a smart-home environment with a HbbTV-enabled television and consumer devices (i.e., smartphones and tablets) through HbbTV v2.0.1. In addition, it offers services such as a dashboard that displays data of smart-home devices or allows the control of these devices using a remote control or voice commands. Besides that, it enables automated features such as auto-pause during smart-home incoming events, such as when the doorbell rings.

However, using HbbTV as the central hub for smart-home device notifications may annoy the user and negatively influence the quality of experience (QoE) when watching video content. The work in [35] evaluates the effects of on-screen notifications on the user’s QoE. It presented a subjective quality evaluation research including 30 participants. These individuals were observed while watching television, with the eye’s gaze direction used to determine the users’ attention to notifications. The most significant finding is that while pop-ups are generally accepted, pop-ups accompanied by sound are viewed as more annoying.

Privacy is an inherent concern when exchanging data over the Internet, and the HbbTV systems are no different. The works in [36,37] discuss some issues related to HbbTV consumers’ privacy and security. For example, they have shown how broadcasters and neighbors can use HbbTV to track users. Besides that, they have detected some suspicious behaviors: some tracking scripts were found, which are responsible for collecting consumer data to obtain more accurate profiles; cookies with unique IDs and a lifetime of one or more years have been found; some channels transfer these data using the HTTP protocol, which is not encrypted, allowing the data to be easily intercepted and read by a third party.

### 3.5. HybridCast

HybridCast is a hybrid television system built using the HTML5 application service platform. The HybridCast’s technical specification was defined in March 2013 by IPTV Forum Japan (IPTVFJ). This specification defines a system model, application model, application control signals, and receiver behaviors. In addition, it also establishes a mechanism for integration with second-screen devices, such as smartphones and tablets.

The following two documents define the interactivity of HybridCast:The IPTVFJ STD-0010 (Integrated broadcast-broadband system specifications) defines the system model, application model, application control signals, transport protocols, VoD, and receiver functions [24];IPTVFJ STD-0011 (HTML5 browser specification) defines HTML application structure, behavior and syntax of elements, and additional objects and APIs [38].

Figure 4 depicts the overall system concept for a HybridCast receiver. A HybridCast terminal can receive DTV signals through an ISDB transmission, where it receives all linear content, AIT data and application data. Through the broadband interface, a receiver can access non-linear content available on the Internet, and the user can also download interactive applications stored in online repositories. The HybridCast system allows third parties other than broadcasters to enter the service chain as application developers and distributors. Like HbbTV, the HybridCast also provides a synchronization manager component, which allows the synchronization of the non-linear content delivered through the Internet to linear content received via the broadcast interface.

A typical Japanese digital television receiver can support data-broadcasting services through the Broadcast Markup Language (BML). A HybridCast receiver includes both a BML and an HTML5 browser. The receiver has four layers: the application, the browser, the middleware, and the hardware [39]. The middleware layer is responsible for handling the linear broadcasting resources. Finally, an application manager controls apps, allowing applications to start and stop.

Like the HbbTV system, HybridCast supports the Companion Screen, and several works propose various services using this functionality. In [40], the authors proposed a solution that tries to improve the experience for deaf people, delivering Japanese sign language animations (computer graphic) content using the Internet displaying it on a second screen in synchronized with the broadcast content. It can be useful, especially during natural disasters such as earthquakes and tsunamis that are common in Japan.

The work in [41] has demonstrated a seamless integration between broadcast content and a calendar application. It enables users to register program schedules and view television shows easily. For example, users can use a smartphone calendar application to schedule TV programs and watch them on TV using a simple click button action.

A mobile-centric integration approach between TV and smartphone was proposed in [42]. The Companion Screen devices work as connectors and data routers over TVs and mobile applications, helping users transit smoothly from mobile to broadcast services. They have extended the HybridCast system to bridge broadcast, Internet, and real-life services using Companion Screen. This architecture allows various mobile applications to access broadcast services through a simple one-tap action smoothly. In addition, it is not restricted to smartphones. For example, it can be implemented in smart speakers, smartwatches, and other devices, enhancing the integration between broadcast and IoT services.

Besides integrating other devices, the IBB proved to be an excellent tool for extensive Internet-based surveys, especially when one of the target audiences is elderly individuals unfamiliar with other connected devices. For example, the authors in [43] conducted a survey on sleep duration and insomnia associated with the NHK TV using the HybridCast TV system.

### 3.6. ATSC 3.0

In November 2017, the United States Federal Communications Commission (FCC) approved the ATSC 3.0 [44], an optional new digital television standard that addressed the ATSC 1.0’s mobile and Internet issues (ATSC 2.0 was released and handled some of these issues, but was never adopted in practice and broadcasters seems to move directly to ATSC 3.0 specification). The ATSC 3.0 is based on the Internet Protocol (IP) set of standards and employs transmission technology suitable for mobile reception. These two capabilities and additional improvements such as substantially enhanced video (e.g., 4K, HDR) and audio (e.g., Dolby Atmos, multi-channel sound).

The introduction of the ATSC 3.0, commonly known as the “Next Gen TV,” is similar to that of the ATSC 1.0, which is not backward-compatible. The ATSC 3.0 will not work on ATSC 1.0 digital television sets, just as ATSC 1.0 did not operate on analog television sets. The FCC ruled that the industry may implement ATSC 3.0 on a station-by-station, market-by-market basis without a defined timeframe, unlike the transition to ATSC 1.0, which had a particular FCC-mandated cutoff deadline. As a stopgap measure, the FCC compelled broadcasters initiating ATSC 3.0 services to continue to offer their principal program service to consumers in ATSC 1.0 format for five years after the FCC approves the new standard, i.e., until November 2022.

In addition to providing standard linear programming, ATSC 3.0 also supports enhanced linear programming and application-based services. Enhanced linear programming can have video, audio, and caption streams selected and synchronized for presentation at the receiver. These programming services can also support interactive games or targeted ad insertion. Furthermore, application-based services are supported, where an application serves as a launching point of the service consumed within the application. An example of such a service could be an on-demand service that allows viewers to browse a library of on-demand content and play selected items.

The ATSC 3.0 receiver is built following a layered architecture, which has several benefits, such as upgradeability and extensibility. Figure 5 shows the overall ATSC 3.0 standard, where the application layer is responsible for providing the environment necessary for the proper operation of a broadcaster application. The files associated with a broadcaster application are supplied in route packages in the Application Context Cache. The receiver’s Web Server makes the broadcaster application’s pages and resources available to the HTML5 browser. Starting an application in an ATSC 3.0 receiver operates similarly to launching a standard Web application in a regular web browser, with no specific behavior or interaction from the receiver.

A broadcaster application is a collection of HTML5, JavaScript, CSS3, XML, images, and multimedia files sent individually or as a package. The application runs within a W3C-compliant web browser, accessing some of the receiver’s graphical components to display the user interface or accessing some of the receiver’s resources or information. When a broadcaster application needs access to resources such as program information, it uses Web Socket Server to request it, using the JavaScript Object Notation–Remote Procedure Call (JSON-RPC) messages, defined in the A/344 specification [45]. These JSON-RPC messages enable the broadcaster application to query information obtained in the receiver, listen to notifications through broadcast signaling, and request the execution of activities not accessible via standard JavaScript APIs.

Additionally, the ATSC 3.0 standard supports the companion device standard (A/338), which defines how a separated connected device, referred to as the Companion Screen, interacts with the receiver (the main device). With the Companion Device Manager APIs defined in the A/338 standard, a broadcaster application may identify and start companion device applications available on the companion device. This API also provides a way for the broadcaster application to acquire Web Socket Service endpoints, enabling companion device apps and the broadcaster application to communicate. The broadcaster application also supports multiple connections.

Emergency alert wake-up is one of the features included in the ATSC 3.0 standard for enhanced emergency communication. Moreover, reliable broadcast networks may provide emergency information, such as danger alerts and critical information. Some recent works supported by the Korean government take advantage of these ATSC 3.0 features to handle emergency alerts. For example, the work in [46] presented a solution of an emergency alert gateway that can provide emergency alerts and disaster information via TV and companion devices at home and send emergency alert messages to the digital display in public facilities. Another work [47] proposes a technique using a deep learning-based algorithm to detect the first bootstrap alert signal even in a low signal-to-noise ratio (SNR) environment.

Regarding the Targeted Advertising (TA) feature, the ATSC 3.0 is still in an early stage, unlike HbbTV, which has this functionality released through the official ETSI specification [33]. The work in [48] proposes an application-based TA system for ATSC 3.0 receivers. The system delivers events to control the targeted ads in real time. The application in the receiver interprets this information to define the ad behavior exposure according to the user preferences and viewing context.

### 3.7. DTV Play (Ginga-D)

In 2016, the Brazilian Association of Technical Standards (ABNT) standardized the Ginga Profile C (Ginga-C), which met some IBB requirements. For example, Ginga-C provides presentation support to a secondary screen and access to remote content using the HTTP protocol. Moreover, the Ginga-C provides an application catalog, where users can select and launch interactive applications. Section 2.3 gives a more detailed description of the Ginga-C profile. However, a fully compliant IBB standard was proposed only in 2018 with a revision to the ABNT NBR 15606 standard, called Ginga Profile D (Ginga-D), also known as DTV Play. It defines the Ginga-HTML5, which supports HTML5 applications (NBR 15606-10) [49] and an HTTP server with REST API, which is called Ginga-CC WebServices (NBR 15606-11) [50]. Besides that, Ginga-D also defines support for streaming content through MPEG dynamic adaptive streaming over HTTP (MPEG-DASH) and HTTP Live Streaming (HLS).

Figure 6 depicts the overall description of the DTV Play receiver architecture. The Ginga Common Core (Ginga-CC) is responsible for handling the display of the various media objects that make up an application, such as JPEG, MPEG-4, MP3, and GIF, among other formats. The Ginga-CC also controls the data transmitted by broadcast and the return channel (e.g., TCP/IP and SMS for mobile devices) to obtain and transmit data on demand. For a better integration between a Ginga receiver and other connected devices (e.g., smartphones. tablets, IoT), ABNT proposed the Ginga-CC WebServices as a new component for implementation in the Ginga-D standard. According to the norm ABNT NBR 15606-11, the Ginga-CC WebServices provides a REST API for access to broadcast information from any application (including interactive applications or native applications), both from the receiver itself and from other devices present in the home environment [50]. This WebServices API is divided into the following groups:The *Client Identification and Authorization* is responsible for establishing the initial link between a client (local application or on a remote device) and the Ginga-CC WebServices. Only once this link is established can these clients use the other APIs.The *Access of DTV context* provides access to the list of available DTV services (TV channels) and their status on the receiver. It also allows switching TV channels.The *Communication Interface to Ginga Execution Environment* allows listing, accessing, and controlling the interactive application available in a specific service.The *Access to SI/PSI Tables and Metadata* provides access to essential tables sent by broadcasters in the transport stream. For example, the Program Association Table (PAT) and the Electronic Program Guide (EPG).The *Access to Multimedia Content from the Broadcaster* allows applications to access the streams and media players running on the DTV receiver.The *Platform and Home Environment Integration* allows for checking the platform characteristics, such as screen size and resolution supported, and the DRM systems supported by the receiver.The *Broadcaster Security Context* provides a binding token from the broadcasters, which the receivers process to create a secure connection between the DTV service environment and apps.

Recent works have proposed extending user interaction possibilities with the Ginga-D standard. For example, in [51], an extension to the Ginga middleware was proposed to allow users to interact with graphical elements of the application using only eye-tracking, creating a new event type in Ginga-NCL, called EyeGaze. Meanwhile, in [52], a framework was proposed to extend the Ginga-NCL, allowing the integration of other interaction modes, such as voice interaction, gestures, and face recognition. In addition, the work in [53] proposed a framework implemented in Lua to integrate Internet-of-Things (IoT) devices to Ginga receivers through the MQTT protocol (publish–subscribe network protocol for IoT). For example, an interactive application can turn on or off a light bulb connected to the network. Moreover, an application can query proximity or humidity sensor data and act based on it.

The work in [54] proposed an architecture to enable the integration of virtual-reality technologies as a second-screen device in interactive multimedia presentations in TV receivers. Although the TV presents traditional content from an application, users with Head-Mounted Displays (HMD) can watch an immersive format, e.g., 360-degree videos, as additional content. The communication between the receiver and the HMD is made using Ginga-CC WebServices, allowing the exchange of messages between them and sending the 360 scenes to be displayed.

Some works perform the integration between Ginga and smart-home systems easier. For example, the works in [55], and [56] proposed a new API based on the Open Services Gateway initiative (OSGi) that allows data from the interactive DTV to be exchanged with home automation devices, including sensors and mobile devices. Furthermore, the work in [57] designed and implemented shows some use cases for this integration. For instance, an image-capturing apparatus can create a camera-monitoring environment in an iDTV-centered network.

### 3.8. IBB Systems Adoption

As described in the previous sections, there are four emerging IBB solutions worldwide: HbbTV in Europe and some countries in Africa, Asia, and Oceania; HybridCast in Japan; ATSC 3.0 in the United States; Ginga-C and DTV Play in Latin America. Figure 7 shows the distribution of IBB systems around the world.

HbbTV is the most widespread IBB system globally because of its compatibility with the DVB DTV system, which already has a large installed base in the African, Asian and European continents. In addition, it constantly updates its specifications, where features not in use are removed, and issues found during the implementation of receivers are fixed. It also has a robust test suite that helps maintain quality and uniformity across manufacturers.

Currently, HbbTV has great acceptance in the markets where it is found. For example, one of the first to join the HbbTV standard, the German market has sold approximately 40 million smart TVs since 2012. About 90% of them support the HbbTV system [58].

HybridCast is compatible with the Japanese ISDB-T DTV standard, thus limited to the Japanese market. However, TV production with HybridCast is growing yearly, emphasizing the years 2020 and 2021 due to the Olympic games [59]. Therefore, the estimation is that smart TVs with embedded HybridCast will reach 75% of the total produced.

The ATSC 3.0 standard, unlike the other IBB systems, required a switch from the old system to the new standard due to non-compatibility. The first significant deployments of ATSC 3.0 occurred in South Korea, with the country’s major television networks launching terrestrial ATSC 3.0 services in May 2017 in preparation for the 2018 Winter Olympics. In November 2017, the United States Federal Communications Commission (FCC) approved regulations allowing broadcast stations to offer ATSC 3.0 services voluntarily. With a presence in an increasing number of markets in the USA, it is estimated that the ATSC 3.0 standard already has about 50% of the American viewers covering 41 cities [60]. Furthermore, it is estimated that about 30% of the TVs manufactured in the US market by 2024 will come with support to the ASTC 3.0 standard [61].

Regarding Ginga Profile C, which the International Telecommunication Union has already recognized as an IBB system for meeting most of the requirements defined in J.205, it has been consolidated in several Latin American countries since the 2000s [62]. On the other hand, we have Profile D, or DTV Play, which introduced support for HTML5 applications and WebServices and is being deployed only in Brazil until 2021. As of 2021, all smart TVs should have this new device. In 2021, 30% of the TVs sold in Brazil had DTV Play on board. By 2023, the perspective is that 90% will have the technology [22,63].

## 4. What Is Next? What Are the Current Technology Challenges?

The IBB systems have significantly improved DTV interactivity, enabling seamless integration of personal mobile devices with DTV receivers. However, these advancements have introduced new challenges, which we will discuss in this section.

The presence of broadband interfaces in IBB receivers has opened the way to access a vast amount of content on the Internet. Still, there is not yet a robust content recommendation mechanism that makes it possible to bring personalized content to the viewer of these devices. Even more, if we consider that TVs are social devices, where different people (with other preferences) at different times may consume content on the same device or even together, identifying the viewers and suggesting content for them is a significant challenge.

Furthermore, private and sensitive user data, such as the history of channels watched and access credentials to interactive applications, can be exposed and misused by third parties on IBB systems. Some works have already shown the presence of these vulnerabilities in systems such as HbbTV [36,37]. Therefore, identifying these problems and mitigating them to keep these devices safe and reliable becomes one of the main pillars for the success of IBB systems.

In traditional 2000s DTV systems, the broadcasters were always responsible for defining which interactive applications the viewers would have access to. This has changed with IBB systems since some systems already allow third-party companies to offer apps to be consumed, such as HbbTV and HybridCast. In addition, to create a more attractive environment for these other developers, it is necessary to standardize these Application Repositories similarly to what is currently done in the smartphone application stores.

The adoption of HTML5 by all IBB systems as a technology for creating applications has made the developer’s life easier since there is already a lot of documentation and frameworks available. However, there are still differences in developing interactive applications for each of these systems, mainly regarding integrating the DTV and Second-Screen APIs that change from one system to another. Nevertheless, there is a clear path for tools to evolve in this direction, allowing developers to generate artifacts for these different IBB systems transparently.

Emergent technologies may be considered a challenge when considering its integration to television in order to provide a new experience. Regarding the Brazilian Digital TV standard, a TV 3.0 call for proposals has been opened [64] for Phase 1 of the project containing requirements and specifications that should be attended. Talking specifically about the application coding, Video on Demand (VoD), push VoD and catch-up TV services are required, along with compatibility with immersive A/V technologies, such as videos with 3 Degrees of Freedom (3DoF) and 6 Degrees of Freedom (6Dof), to support Virtual Reality (VR), Augmented Reality (AR) and Extended Reality (XR). For A/V and captions coding, the requirements include a video resolution of up to 7680 × 4320 (8K), HDR, Wide Color Gamut, progressive frame rate up to 120 fps, Three-Dimensional (3D) audio and sign language video stream. All these are challenges for the next generation of television.

On the other hand, HbbTV in its new release 2.0 has focused on user experience, seamless viewing of video content across TV, smartphones, PCs and tablets, innovative companion applications that enhance the TV experience with detailed program info, voting, play to screen and other use cases, standardized delivery of ultra HD content with HEVC, improved accessibility of services with better support for subtitles in multiple languages, access to broadcast content captured to local storage in the receiver and support for consumer privacy [65].

By removing the bottleneck of network speed and latency, it is possible to add a huge computational capability to any connected device by running computation demanding tasks on the cloud. It would be interesting to see how television and other devices will explore these possibilities in the future. It is possible to see even thinner devices with very low power consumption executing very complex rendering tasks. Even the interactivity module could be provided as a service being hosted elsewhere, resolving several compatibility issues as all devices could use the same software.

From the next generation of broadcast and broadband systems, we expect to see an even better integration with emerging telecommunication technology, such as 5G networks, as well as the aggregation of immersive experiences such as virtual reality or expanded reality together with residential automation systems where a user can ask a digital assistant to execute an operation directly on the television. In other words, we expect to see several systems working together performing in unison to provide the best experience and convenience to the user, providing a really customized feel everywhere, and the television itself will be a piece of this whole and the whole greater than the sum of the parts.

## 5. Conclusions

This paper shows the evolution of TV interactivity systems in recent decades. We approach the first interactive systems such as MHP and ACAP that emerged in the early 2000s in a scenario where TVs (or set-top boxes) were very resource-constrained (e.g., low memory and CPU available). Moreover, they had an inefficient feedback channel, which made the interaction between broadcasters and viewers difficult. As a result, these first systems became obsolete quickly, mainly due to the emergence of smartphones and smart TVs that dominated the scene in the 2000s, leading to a drastic change to how users consumed and interacted with content (e.g., video on demand, social networks, etc.).

We have shown how this new scenario motivated the creation of study groups around the world, which led to the creation of the new generation of interactivity systems, known as IBB, that allowed the creation of new services that took advantage of the best of broadcast and broadband worlds. It was pointed out new perspectives for the next generation of TV experience, covering aspects related to applications, devices, and tools. We can conclude that these new systems open new opportunities for developing new interactivity services for TVs in the following years, allowing a greater integration between broadcasters and viewers.

## Figures and Tables

**Figure 1 sensors-22-06542-f001:**
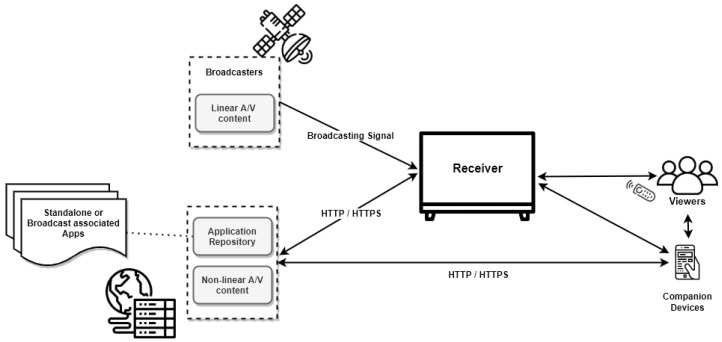
Integrated Broadcast Broadband (IBB) architecture entities.

**Figure 2 sensors-22-06542-f002:**
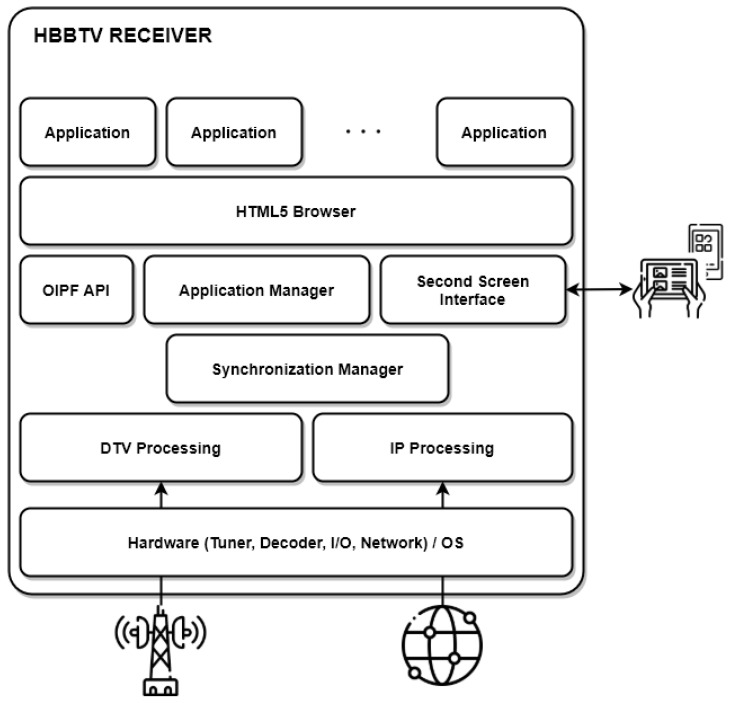
Overall HbbTV Receiver Architecture.

**Figure 3 sensors-22-06542-f003:**
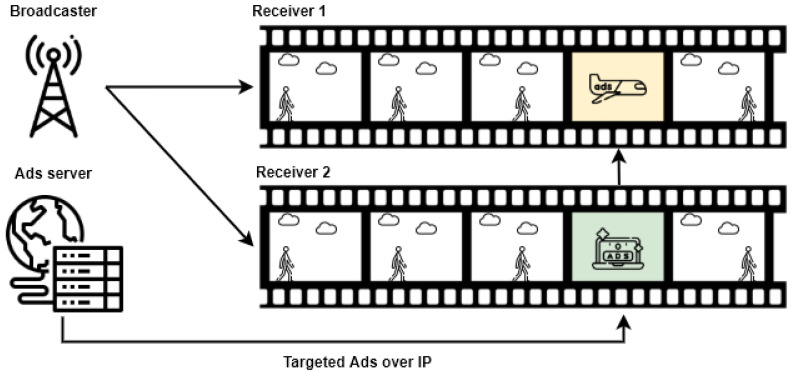
HbbTV targeted ads operation.

**Figure 4 sensors-22-06542-f004:**
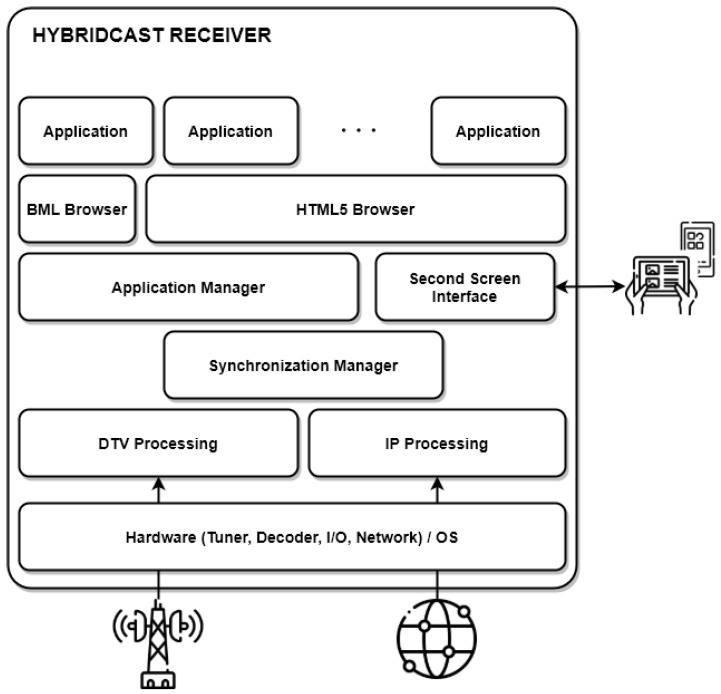
Overall HybridCast Receiver Architecture.

**Figure 5 sensors-22-06542-f005:**
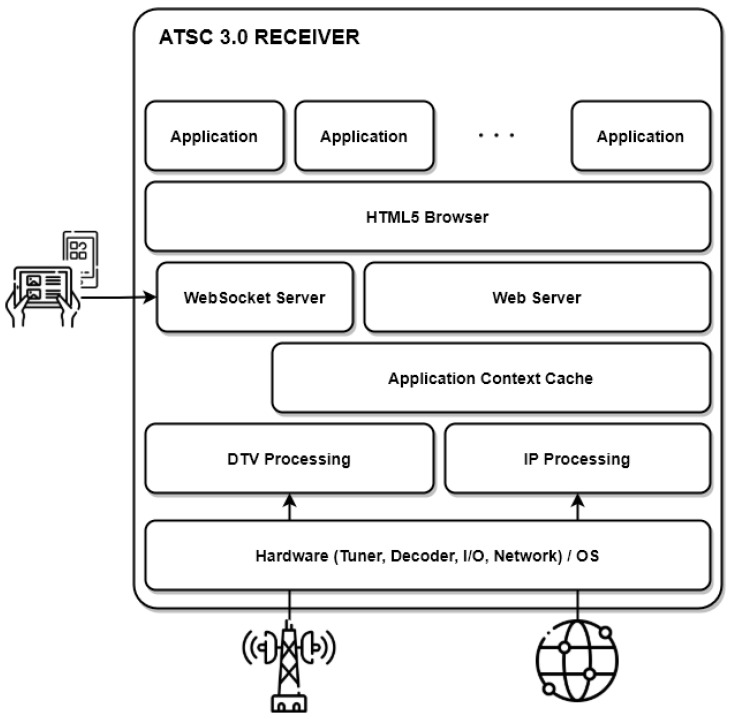
Overall ATSC 3.0 Receiver Architecture.

**Figure 6 sensors-22-06542-f006:**
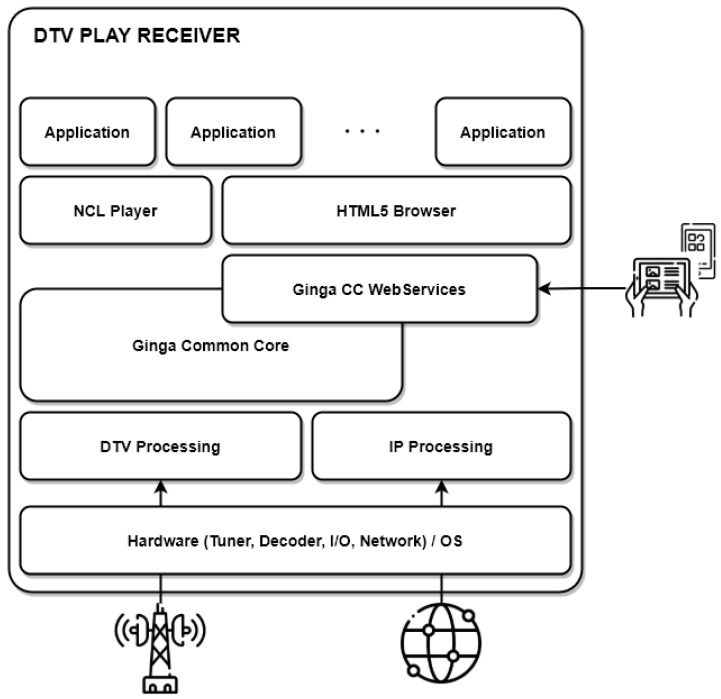
Overall DTV Play Receiver Architecture.

**Figure 7 sensors-22-06542-f007:**
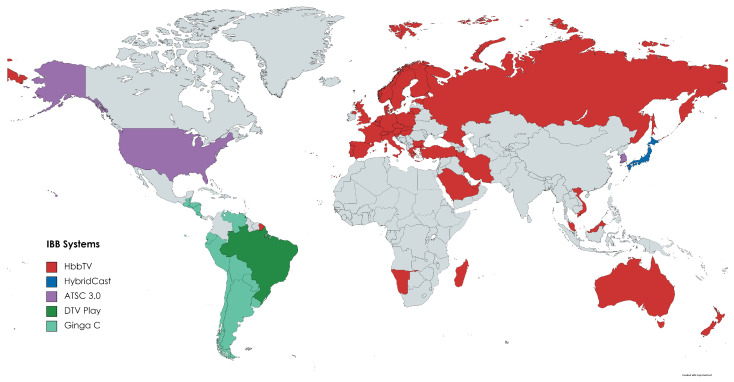
IBB systems deployments in 2021.

**Table 1 sensors-22-06542-t001:** Early ITV middlewares comparison.

Middlewares	MHP	DASE	ACAP	OCAP	BML	GEM	Ginga
Procedural Applications	Java TV Xlet						
Declarative Applications	HTML						NCL/Lua and HTML
Communication Protocol	TCP/IP and UDP/IP						TCP/IP and SMS

**Table 2 sensors-22-06542-t002:** IBB interactive supported features.

IBB Features	HbbTV	HybridCast	ATSC 3.0	Ginga-C	DTV Play (Ginga-D)
Supported languages	HTML5	HTML5	HTML5	NCL/Lua	NCL/Lua and HTML5
Support second screen	Yes	Yes	Yes	Limited	Yes
Video stream support	MPEG-DASH	MPEG-DASH and HLS	MPEG-DASH	No	MPEG-DASH and HLS
Broadcast content access	Yes	Yes	Yes	No	Yes
Robust test suite	Yes	Yes	No	No	No
Third party as content provider	Yes	Yes	No	No	No
Targeted ads ^1^	Yes	Yes	Yes	No	No

^1^ Targeted ads inserted in linear broadcasting content.

## Data Availability

Not applicable.

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
