# Peer review of "A Survey of Digital Television Interactivity Technologies"

_sensors, 2022, doi:10.3390/s22176542_

Round 1

Reviewer 1 Report

The authors raise a very interesting issue related to DigitalTV. This is a very wide topic, with many scientific, research and social engineering aspects. In this case, the authors were tempted to collect information on the most popular standards related to Digital TV Interactive Technologies. The authors presented the history and evolution of the standards appearing recently, briefly commenting on their pros and cons. They cite appropriate literature sources. They also describe the latest and the most popular. In my opinion, the presentation method and the content itself, although it is primarily a gathering of generally available knowledge, it can constitute an interesting reading for people interested in the subject, although not professionals. For professionals in this field, it is difficult to find in this article some particularly valuable information.

The fourth chapter is important. In my opinion, the authors should develop it because it is de facto cherry on the top of their work.

This is a typical technology review based on available sources. I think the type of work should be changed to "review" instead of "article".

After the expansion of Chapter 4, "Review" may be published.

Reviewer 2 Report

The paper is written in proper English, it is pleasant to read. The text is informative, followed by appropriate citations. It talks not only about broadcasting standards worldwide, but also related reception equipment such as STBs, applications, etc. Overall, it provides a good source of information for both professionals, academics, as well as any interested third-party interested in the development of television.

I do recommend this paper to be accepted and published in the Journal. Surely, I will be one of the first readers to acquaint with it online. I will surely share it with my Students at the University.

Suggestions and comments that could further raise the quality of this paper:

Personally, I tend to write Internet with a Capital I, just like Bluetooth, as it is the name of a technology.

Figure 1 could be larger in size, with larger fonts, making it easier to read and interpret. Furthermore, it would look better at higher resolution, dpi and/or different file format, so that the lines look sharper.

It would look interesting to prepare a summary of pros and cons between cable, terrestrial and satellite TV.

It would look also interesting to have a map of the world with color-marked standards present in each country/continent, or just the country of origin of respective standards.

In a survey, it would be advisable to prepare a table summing up all discussed broadcasting standards, with similarities and differences between them.

Minor editorial and formatting issues, e.g., multiple (unnecessary) space signs between subsequent words, etc.

Round 2

Reviewer 1 Report

The authors fulfilled my postulate. In my opinion, you can publish it as a review.